# Effect and Mechanism of Pitch Coating on the Rate Performance Improvement of Lithium-Ion Batteries

**DOI:** 10.3390/ma15134713

**Published:** 2022-07-05

**Authors:** Bo-Ra Kim, Ji-Hong Kim, Ji-Sun Im

**Affiliations:** 1C1 Gas & Carbon Convergent Research Center, Korea Research Institute of Chemical Technology (KRICT), Daejeon 34114, Korea; bora4660@krict.re.kr (B.-R.K.); pic10@krict.re.kr (J.-H.K.); 2Department of Chemical Engineering and Applied Chemistry, Chungnam National University, Daejeon 34134, Korea; 3Advanced Materials and Chemical Engineering, University of Science and Technology (UST), Daejeon 34113, Korea

**Keywords:** pitch, carbon coating, graphite, lithium-ion batteries

## Abstract

This study evaluated the effect of pitch coating on graphite anode materials used in lithium-ion batteries and investigated the mechanism whereby pitch coating improves the electrochemical properties. The FG (flake graphite) and pitch were mixed in weight ratios of 95:5–80:20. The mixture was pressed and prepared into a block form. Additionally, heat treatment was performed at 900 °C for 1 h and pulverized in the size range of 10–25 μm. The results showed that the particles of uniform pitch-coated graphite became more spherical. However, when the pitch is added excessively, pitch aggregation occurs rather than a thicker coating, indicating a nonuniform particle shape. Pitch has a randomly oriented structure and a small crystal size. Therefore, pitch serves as a lithium-ion diffusion pathway, resulting in an improved rate of performance. Notably, the uniform pitch-coated graphite exhibited an outstanding rate of performance owing to the relieving of particle orientation in the electrode rolling process. During the rolling process, the particles are oriented perpendicular to the lithium-ion diffusion pathway, making it difficult for the lithium ions to diffuse. Adding an excessive amount of pitch was found to deteriorate the rate of performance. Pitch aggregation increased the interfacial resistance by forming a heterogeneous surface.

## 1. Introduction

Global warming and environmental pollution have led to abnormal changes in weather, and regulations to reduce carbon emissions have been strengthened. The adoption of electric vehicles (EVs) is emerging as a new direction to reduce our reliance on fossil fuels, and the development of batteries, which are energy storage devices for EVs, has become important, particularly lithium-ion batteries (LIBs). Graphite is widely used as an anode material in LIBs, given its low reduction voltage (~0.2 V vs. Li/Li^+^), which is close to the reduction voltage of lithium metal, and negligible volume change during the intercalation/deintercalation of lithium ions [1,2], yielding an excellent cyclability. However, since the pathway of lithium ions is limited, the charge/discharge efficiency at high rates is low [3], and the power density is insufficient for the latest batteries. The rate performance of graphite anode materials can be improved by graphite spheronization, surface doping, and carbon coating [4,5,6,7]. Among them, coating with pitch-derived carbon is a cost-effective method. Pitch is a carbon source of the residue obtained from the heat treatment and distillation of petroleum fractions or coal tar. It is a solid at room temperature, comprising a complex mixture of numerous predominantly aromatic hydrocarbons and exhibiting a broad softening range instead of a defined melting temperature [8]. The raw materials in pitch can be petrochemical by-products such as pyrolysis fuel oil, fluid catalytic cracking-decant oil, and vacuum residue, which are cost-effective, abundant, and high in carbon content with an aromatic structure [9]. Moreover, pitch coating has the advantage of imparting sufficient mechanical strength to moderate the volumetric expansion [10] and high conductivity [11]. Hence, it has attracted considerable interest as a carbon coating source [12,13,14,15,16].

A carbon coating is applied to increase the initial coulombic efficiency (ICE) by reducing the initial irreversible capacity through the formation of a uniform surface, and the amorphous carbon on the surface provides a lithium diffusion pathway to improve the rate performance [17,18]. However, the reason behind this characteristic improvement and the adverse effects of adding an excessive amount of carbon have not been clearly explained.

In this study, the physical properties and electrochemical performance of electrodes made of graphite, suitable pitch-coated graphite, and excessive pitch-coated graphite were analyzed and compared. Finally, the mechanism whereby the carbon coating improves the electrochemical properties was thoroughly examined.

## 2. Experimental

### 2.1. Sample Preparation

Pitch-coated graphite was prepared from natural flake graphite (FG, D50 15 μm) and pitch (petroleum, softening point: 255 °C, D50: 4 μm), a carbon coating source. The FG and pitch were mixed in weight ratios of 95:5, 90:10, 85:15, and 80:20 via centrifugal mixing for 30 min. The mixture was pressed using a hot press at 315 °C for 30 min and prepared into a block form. Given its softening point, the pitch can flow down during the heat treatment and cause a mass gradient. To prevent this, the FG and pitch were mixed and then pressed to prepare the block form. Heat treatment was performed at 900 °C for 1 h, and the prepared sample was pulverized using a mortar and classified in the size range of 10–25 μm. The carbonized samples were named CFGX, where X represents the weight percent of the pitch. To evaluate the amount of pitch actually contained, it was calculated using the density difference as shown in Appendix A. Graphite and pitch have a density difference that occurs in the crystal structure, and the content of pitch was calculated using the true density of each sample. CFG5, CFG10, CFG15, and CFG20 were 4.10%, 7.03%, 8.92%, and 13.00%, respectively, which is different from the initial addition amount. This is because the residual amount of pitch after heat treatment is different. Appendix A shows the thermal analysis estimated under the same conditions (5 °C/min, 25–900 °C, N_2_ atmosphere) as the heat treatment after coating, and the carbon yield was 53.53%. Considering this, it can be seen that the actual pitch content was reliable results.

### 2.2. Characterization

True density was measured by a He pycnometer through Micro meritics AccuPyc II 1340. Scanning electron microscopy (SEM) was performed to observe the particle morphology of the prepared samples, and the particle size distribution was analyzed using Microtrac Bluewave (laser diffraction method) to determine the degree of particle aggregation. Surface crystallinity was investigated using Raman spectroscopy, and the average Ad/Ag (area defect/area graphite) value and standard deviation were obtained by performing the analysis thrice for each sample to determine the particle surface uniformity. The wavelength of the laser used for Raman spectroscopy was 532.04 nm, and the silt width was set to 50 mm. A powder resistivity measurement system was used to determine the electronic conductivity when the prepared sample was roll pressed, and the sample packing density depending on the load level was found to investigate the effect of pitch coating after roll pressing. Packing density measured using powder resistivity measurement is different from true density. True density is a solid density, and the packing density is defined as the ratio of the only solid volume to the total sample volume [19,20]. The powder conductivity values were measured using a powder resistivity measurement system under load between 400 and 2000 kgf (Hantech, Republic of Korea).

### 2.3. Electrochemical Measurements

Electrodes were prepared to evaluate the electrochemical properties, and carboxyl methyl cellulose (CMC) and styrene-butadiene rubber (SBR) were used as binders. A slurry was prepared by mixing 95 wt% active material, 2.5 wt% CMC, and 2.5 wt% SBR, and coated on a copper foil (as a current collector) using a doctor blade. Subsequently, the electrodes were dried in a vacuum oven at 80 °C for 12 h. The dried electrodes were punched into disk (∅ = 13.5 mm) forms; the material loading level was 4.5–5 mg cm^−2^. The electrodes were roll-pressed to ensure particle contact and to constantly adjust the porosity (~30%). The 2032-coin cell was assembled in an argon-filled glove box to avoid any effect of moisture and oxygen. The electrolyte was 1 M LiPF6 dissolved in a mixture of ethylene carbonate (EC) and diethyl carbonate (DEC) (1:1 by volume), and polyethylene (PE) was used as a separator. The fabricated cell was aged for 24 h to allow sufficient contact between the active material in the electrode and the electrolyte. Electrochemical tests were conducted using a WonATech WBCS3000 (Seoul, Republic of Korea).

The galvanostatic charge and discharge tests were conducted in a potential range of 0.01–1.5 V (versus Li/Li^+^). The coin cells were lithiated at a rate of 0.1 C and delithiated at various rates ranging from 0.1 C to 5 C to estimate the rate of performance; the current density at 1 C was 372 mA g^−1^. Electrochemical impedance spectroscopy (EIS) was performed at an amplitude of 10 mV and in the frequency range of 100 kHz–0.01 Hz. Cyclic voltammetry (CV) was conducted in the range of 0.01–1.5 V (versus Li/Li^+^) at voltage scan rates varying from 0.1 to 2.0 mV s^−1^ to analyze the diffusion coefficient.

## 3. Results and Discussion

### 3.1. Properties

Figure 1 shows the SEM images of FG, CFG5, CFG10, CFG15, and CFG20. The morphology of FG is a mixture of flake type and spherical type, and the surface is rough. CFG5 has a more spherical shape and a more uniform surface than FG. In total, 5 wt% pitch on the graphite surface is thought to have improved the spheroidization and surface uniformity of the particles. Appendix A also confirmed the clean surface and regular spherical particle shape. It provides a lithium-ion diffusion pathway; thus, an enhancement in the rate of performance can be expected. This is further examined in Section 3.3 and Section 3.4. Partially spherical particles can be observed in CFG10; however, pitch aggregation is also confirmed. Aggregation is pronounced with increasing pitch content; in particular, the particle surface in CFG20 is rougher than that in FG. Notably, since a large amount of pitch causes aggregation rather than an increase in the coating thickness, a nonuniform particle morphology and surface are formed; hence, it is important to add an appropriate amount of pitch. In addition, pitch aggregation did not lead to the formation of a uniform interface between the particles, which may increase the interfacial resistance during lithium-ion intercalation/deintercalation. This is explained further in Section 3.3 and Section 3.4.

To examine particle aggregation, the particle size distribution was measured and shown in Table 1. In the order of FG, CFG5, CFG10, CFG15, and CFG20, the D90 values are 27.43, 28.85, 29.98, 33.16, and 34.01, respectively, which increase with an increase in the amount of pitch added. In particular, the D90 value increases rapidly from CFG15 because the excessive pitch content causes pitch aggregation as shown in the SEM images. The values of FHWM (full width at half maximum) also increase in the order of FG, CFG5, CFG10, CFG15, and CFG20, which implies an increasingly irregular particle size.

The surface area of all samples was measured by the BET equation, and the pitch-added sample had a significantly lower specific surface area than that of the FG. This suggests that carbon coatings make a clean and smooth surface, so it was expected to improve initial coulombic efficiency.

To further investigate the surface uniformity, Raman spectroscopy was used to measure the surfaces of different particles for each sample thrice. The A_D_/A_G_ ratio was calculated to indicate the degree of surface crystallinity, and the standard deviation was determined to compare the surface uniformity, as shown in Table 2. The peak at 1350 cm^−1^ is ascribed to the disordered structure and surface defect of the carbon material and is indicated by I_D_ (intensity of D band) or A_D_ (area of the D band). I_G_ or A_G_ at 1580 cm^−1^ represents graphite structure, high crystallinity, and sp^2^ carbon-carbon bond [21,22,23]. The average value and standard deviation of the A_D_/A_G_ ratio of FG were calculated to be 0.555 and 0.0452, respectively. CFG5, CFG10, CFG15, and CFG20 showed higher average A_D_/A_G_ ratios than FG; this is because the pitch has a more disordered structure than graphite. The standard deviations of CFG5, CFG10, CFG15, and CFG20 were 0.0138, 0.1324, 0.1567, and 0.2838, respectively, which increased with increasing pitch content. Notably, the standard deviation of CFG5 was lower than that of FG, which means that all the surfaces were uniformly coated with the pitch and that excessive pitch addition (10 wt% or more) made the surface less uniform. In particular, the A_D_/A_G_ values of CFG15 and CFG20 were found to be ~2.2 because the pitch was not coated on the graphite surface and was aggregated due to excessive addition. In addition, XRD for FG and CFG5 was conducted to investigate surface crystallinity as shown in Appendix A. The d002 peak (about 26.5 θ), which means graphene interlayer spacing, is represented at 26.46 θ (CFG5) and 26.48 θ (FG) and the intensity of the corresponding peak was weaker, indicating CFG5 had a disordered structure than FG. This suggests that the surface of CFG5 was covered with a pitch layer, improving surface uniformity.

In summary, under 5 wt% pitch addition, the coating on graphite was even, forming a uniform surface, and the particles were more spherical. However, under 10 wt% pitch addition, aggregation started to occur, and under 15 wt% or more, the particles were excessively aggregated, forming a nonuniform particle surface/morphology. Therefore, it is more appropriate to name it “pitch-added graphite” rather than “pitch-coated graphite” with a pitch addition of 10% or more.

### 3.2. Electrochemical Performance

Figure 2 shows the galvanostatic charge (lithiation)/discharge (delithiation) profiles of graphite and pitch-coated graphite in the first cycle; charging is in the CC-CV mode, and discharging is in the CC mode (0.1 C current density). As shown, the first initial coulombic efficiency (ICE) of CFG5 (84.07%) is higher than that of FG (80.49%); however, the ICE values of CFG10, CFG15, and CFG20 are lower than that of FG, as follows: 80.16%, 78.47%, and 76.88%, respectively. As shown in Table 1, CFG5 had a lower surface area and no-particle aggregation. This implies that CFG5 was uniformly coated with the pitch on the graphite surface, which not only formed a uniform surface but also suppressed any side reactions with the electrolyte by avoiding the exposure of the edge plane of the graphite, resulting in reduced charge capacity and increased ICE. However, excessive pitch addition caused aggregation and did not appropriately prevent exposure of the edge plane, but rather resulted in a nonuniform particle surface. In addition, as shown in Appendix A, it can be confirmed that 0.686 wt% of hydrogen ions remained in the pitch-coated surface heat-treated to 900 °C, causing a side reaction with the hydrogen ions and the electrolyte [24], thus reducing the ICE.

To further compare the side reaction with the electrolyte, a differential capacity analysis (dQ/dV) was performed, as shown in Figure 3. Generally, electrolyte decomposition is formed in the first cycle, and the reaction proceeds in the voltage range of 0.5–0.8 V [25,26]. A differential capacity analysis is an electrochemical analysis method that can help identify the reaction voltage of a set voltage [27]. A peak appeared between 0.5 and 1.0 V in the dQ/dV curve of the prepared sample, related to electrolyte decomposition. The corresponding peak of CFG5 is the smallest, increasing in the order of FG, CFG10, CFG15, and CFG20, which implies many side reactions with the electrolyte.

Figure 4 shows the rate performance at various discharge rates and cycling performance. As shown, the discharge capacity retention of FG is 51.90% at 2 C and 9.07% at 5 C compared to its discharge capacity at 0.2 C (4th cycle). In the case of CFG5, the capacity retention values at 2 C/0.2 C and 5 C/0.2 C are 98.93% and 80.58%, which are 1.9 and 8.8 times higher than that of FG, respectively. The improvement in capacity retention could be explained by the isotropic structure of amorphous carbon on graphite. Graphite can intercalate/deintercalate lithium ions only in the edge plane; however, the amorphous structure of the pitch provides a lithium-ion pathway, which enhances the rate performance [18]. The 5 C/0.2 C capacity retention values of CFG10 (68.15%), CFG15 (67.35%), and CFG20 (48.26%) were lower than those of CFG5 because the formation of a nonuniform surface increased the interfacial resistance. However, amorphous carbon has a variety of lithium-ion insertion/desorption sites and a smaller crystal structure than graphite, resulting in an improved rate of performance compared to FG. The reason for the rate enhancement is explained in more detail in Section 3.3 and Section 3.4. All the pitch-added samples represent that the delithiation capacity after the 5 C rate was slightly higher than before. Amorphous carbon shows the structure was arranged, so a slight increase in capacity may occur. Appendix A shows the delithiation capacity of the pyrolyzed pitch increased after the 5 C rate (the delithiation capacity at 0.2 C before the high rate was as follows: 243.88 mAh/g, the delithiation capacity at 0.2 C after the high rate: 244.49 mAh/g).

The cycling stability (delithiation capacity at 0.1 C, 33 cycles)/(delithiation capacity at 0.1 C at 3 cycles) was calculated for each sample, and the results were as follows. FG: 82.36%, CFG5: 100.70%, CFG10: 99.63%, CFG15: 100.48%, CFG20:99.03%. As lithium intercalates and deintercalation was repeated, the graphene layer was exfoliated, and the cycling stability deteriorated. However, when the pitch was added and uniformly coated on the surface (CFG5), excellent stability was confirmed by inhibiting the exfoliation of the graphene layer.

### 3.3. Reason for the Improved Electrochemical Performance of Pitch-Coated Graphite

Many studies related to carbon coating have been conducted to improve the electrochemical properties, such as the initial coulombic efficiency, rate retention, and cycling of anode materials, based on both the intercalation mechanism and the alloying mechanism [7,28,29,30]. In particular, in this study, the rate of performance was significantly improved after carbon coating, which was attributed to two factors. (1) Structure of amorphous carbon, and (2) hindrance to graphite particle orientation (spherical particle effect).

The first factor is that amorphous carbon is disorderly oriented along the c-axis and has a smaller crystal size (L_a_, L_c_) and a larger interlayer distance (d_002_) than graphite. However, graphite has a limited path because lithium intercalation/deintercalation is possible only at the edge plane [31,32]. So, amorphous carbon generally shows remarkable rate performance, and the electrochemical properties of the pyrolyzed pitch were estimated as shown in Appendix A. The pyrolyzed pitch had a plateau at a low potential of 0.05 V as shown in Appendix A, which shows a similar behavior to a soft carbon structure. It has various lithium adsorption/desorption sites, and lithium ions can be stored in irregular hexagonal planes and surface sites, and in clusters between graphene crystals [33]. The rate of performance (5 C/0.2 C) of the pyrolyzed pitch was 48.55%, which had a similar performance to that of CFG20 (48.26%), while it was greatly lower than that of CFG5 (80.58%). This suggests that even when the pitch was simply added, the rate capabilities can be improved due to the structural characteristics of the amorphous carbon, and it further accelerates the enhancement of rate properties when coated uniformly.

Therefore, the pitch coated on the graphite surface provides a lithium-ion diffusion pathway, leading to enhanced rate properties. To confirm this, the charge transfer resistance and lithium-ion diffusion coefficient were estimated using EIS and CV, respectively, as shown in Figure 5. The equivalent circuit in Figure 5a shows that the charge transfer resistance (R_ct_) is related to the intercalation/deintercalation resistance of the lithium ions at the electrode–electrolyte interface [34]; therefore, the higher the R_ct_ value, the more difficult it is to intercalate/deintercalate the lithium ions. In addition, because the EIS analysis is significantly affected by temperature and voltage [35], the measurement was conducted under the same conditions at above 3 V and at room temperature before cycling. The R_ct_ value of FG was the highest and gradually increased in the order of CFG5, CFG10, CFG15, and CFG20. As mentioned above, since the amorphous carbon coated on the graphite surface provides a diffusion pathway for the lithium ions, the resistance decreases with increasing pitch addition. In particular, CFG5, which had the most uniform surface, had the lowest resistance because of the formation of a homogeneous electrolyte–electrode interface. In other words, CFG10, CFG15, and CFG20 showed a lower rate of performance than CFG5 because the nonuniform surface and pitch aggregation caused an increase in the interfacial resistance. In addition, the EIS of FG and CFG5 was measured to compare the diffusion of lithium ion in the electrodes after cycling, as shown in Appendix A. The two semicircles were represented, and it means the resistance of SEI and charge transfer. The R_sei_ and R_ct_ were 12.78 Ω, 38.11 Ω (FG), 4.42 Ω, 27.24 Ω (CFG5). All resistances of CFG5 were reduced, and it was confirmed that the formation of the SEI layer was suppressed due to the reduction of the specific surface area and the uniform surface, and the amorphous carbon coating provided an equivalent diffusion path, which improved the rate performance.

To further understand the electrochemical kinetics, Appendix A shows the CV values at voltage scan rates varying from 0.2 to 2.0 mV s^−1^ in the range of 0.01–1.5 V. In all the samples, the peak current (*I_p_*) increases gradually, and the delithiation peak potential shifts to higher values with increasing scan rate (*v*). Based on the delithiation peak current shown in Appendix A, the relationship between *I_p_* and *v* can be expressed using the following Randles–Sevcik equation [36].
$$I_{p}=2.69\times 10^{5}\;n^{1.5}\;A\;D_{Li}{}^{0.5}\;v^{0.5}\;C_{Li}\;slope=2.69\times 10^{5}\;n^{1.5}\;A\;D_{Li}{}^{0.5}\;C_{Li}$$

In this equation, *I_p_* is the peak current, *n* is the electron transfer number in mole, *A* is the electrode area, *C_Li_* is the intercalated/deintercalated lithium-ion concentration (mol cm^−3^), *v* is the scan rate, and *D_Li_* is the lithium-ion diffusion coefficient where the unit is 10–11 cm^2^ s^−1^. Figure 5b shows the slope calculated from the CV curves. Based on the equation, the lithium-ion diffusion coefficient can be obtained; the steeper the slope, the higher the lithium-ion diffusion coefficient. The lithium-ion diffusion coefficient was calculated by adding the estimated slope and the above factors (*n* = 1, *F* = 96,485 C mo1^−1^, *A* = 1.43 cm^2^, *C* = 0.001 mol cm^−3^). The diffusion coefficient result was similar to the tendency of the charge transfer resistance. The diffusion coefficient of FG showed the lowest value of 0.242, gradually decreasing to 3.354, 3.118, 3.294, and 2.486 in the order of CFG5, CFG10, CFG15, and CFG20, respectively. All the samples to which amorphous carbon was added exhibited a higher lithium-ion diffusion coefficient; this can be attributed to the fact that the structure of amorphous carbon provides a movement pathway for the lithium ions. When the pitch was added excessively, the interfacial resistance increased due to pitch aggregation and nonuniform surface formation, resulting in a rather low lithium diffusion coefficient. 

The other reason for the improvement in the rate of performance of the pitch-coated graphite is as follows. The particles are oriented in a direction perpendicular to the current collector under the force generated during the electrode rolling process [37,38,39]. The pitch coated on the graphite surface imparts a more spherical particle shape because of which the spherical particles are relieved of their orientation [33,40], thus facilitating the diffusion of lithium ions. The density and electronic conductivity were evaluated under loads varying from 400 to 2000 kgf using the powder resistivity measurement system, as shown in Appendix A. In all the samples, the packing density and electronic conductivity increased with increasing load because of the increased contact between the particles and the shortening of the electron pathway [20]. Therefore, we compared the electronic conductivity at the same packing density (1.5 g/cc) at which the electrodes were fabricated, as shown in Table 3. Under the same density, with increasing pitch content, the electronic conductivity decreased, which is inconsistent with the EIS and CV results. This difference is because the electronic conductivity is indicative of electron mobility, whereas electrochemical analyses, such as EIS and CV, are conducted for the diffusion of lithium ions. Nevertheless, notably, the electronic conductivity of CFG5 was similar to that of FG despite the pitch addition, and the addition of a small amount of pitch did not significantly decrease the electronic conductivity. Appendix A shows that the density is lower under the same load as the pitch is added, because the content of the pitch, which was less dense than graphite, increased.

### 3.4. Schematic of Electrode Structure after Pressing Process

Based on these results, Figure 6 shows the schematic of the electrode structure after the pressing process. In the noncoated graphite (FG), during the rolling process, graphite receives a force in the normal direction, resulting in particle orientation in a direction parallel to the copper current collector. The particles are oriented in a direction perpendicular to the diffusion direction of lithium ions; this extends the diffusion pathway, and the diffusion of lithium ions becomes difficult. Hence, surface modification (e.g., carbon coating and spheronization) is essential to increase the rate of performance of graphite electrodes. The uniformly coated pitch on the graphite surface (CFG5) provides a lithium-ion diffusion pathway, helps impart a more spherical particle shape, and suppresses particle orientation during the rolling process. To further confirm this, Appendix A shows the actual and calculated packing densities of CFG5 with respect to the load. The density was calculated as follows:(packing density of FG at load) × (weight percent of FG) + (packing density of pitch at load) × (weight percent of pitch)

The actual density was lower than the calculated one. In graphite, particle orientation occurs in a direction parallel to the current collector when pressed in the normal direction; however, the addition of pitch created more spherical particles, as shown in the SEM images, and the pitch on the surface and the spherical shape helped relieve particle orientation; therefore, the pitch-coated graphite was less compressed by the pressing force. In other words, the difference between the actual and calculated packing densities was due to the relaxation of particle orientation. In the case of graphite coated with an excessive amount of pitch (i.e., CFG10, CFG15, and CFG20), the pitch made it possible to suppress particle orientation, because of which the actual density was lower than the calculated one, as shown in Appendix A. However, the interfacial resistance increased due to the nonuniform surface and pitch aggregation, which deteriorated the rate performance compared to that of the uniform pitch-coated graphite.

## 4. Conclusions

The physical and electrochemical properties of electrodes made of graphite (FG), uniform pitch-coated graphite (CFG5), and excessive pitch-coated graphite (CFG10, CFG15, and CFG20) were analyzed and compared. A difference in the rate of performance was found, and the reason for this was investigated. The following conclusions can be drawn:(1)Graphite (FG) exhibited particle orientation during the rolling process, and since the particles were oriented perpendicular to the lithium-ion diffusion direction, the pathway of lithium ions was extended, and intercalation/deintercalation was limited, resulting in a deteriorated rate of performance of 9.07% (5 C-rate/0.2 C-rate);(2)In the case of uniform pitch-coated graphite (CFG5), the amorphous carbon with a randomly oriented structure and small crystal size provided a lithium diffusion pathway on the surface. In addition, the pitch made the particles more spherical and helped relieve the particle orientation. The result was a superior rate performance of 80.58% (5 C-rate/0.2 C-rate), an improvement of more than eight times compared to that of FG;(3)Excessive pitch addition resulted in pitch aggregation rather than a thicker coating. Such an aggregation caused a nonuniform particle surface and shape, which increased the interfacial resistance and charge transfer resistance and decreased the rate of performance. Nevertheless, the rate performance was improved compared to that of graphite, which was attributed to the amorphous structure of the pitch and relaxation of particle orientation by some of the coated pitches.

## Figures and Tables

**Figure 1 materials-15-04713-f001:**
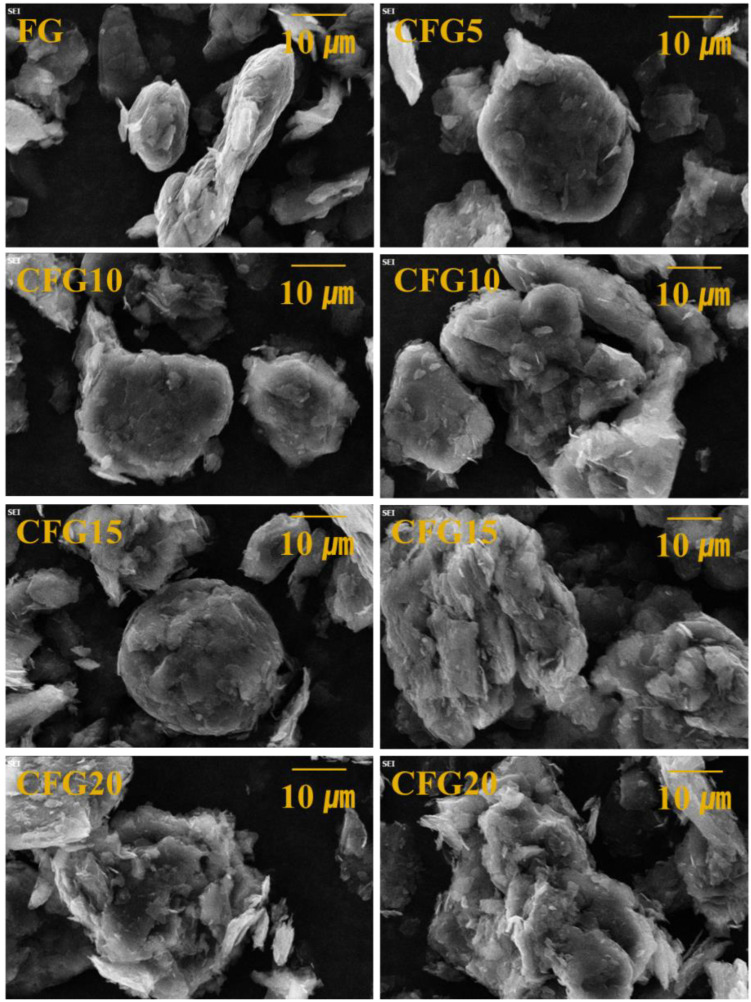
SEM images of FG, CFG5, CFG10, CFG15, and CFG20.

**Figure 2 materials-15-04713-f002:**
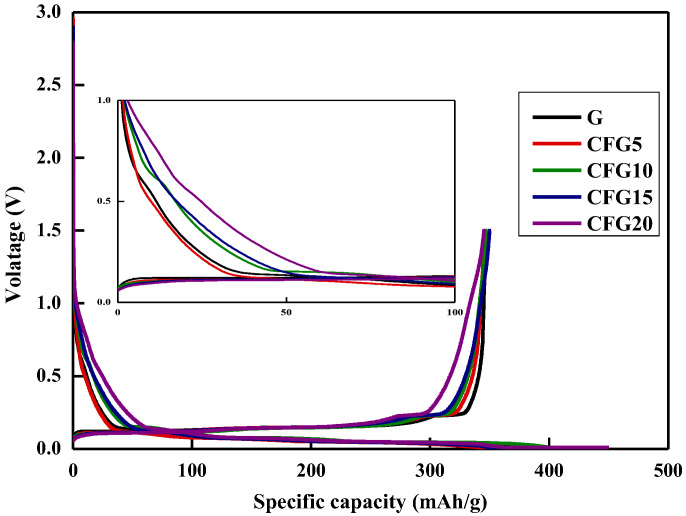
Galvanostatic charge/discharge (GDC) at 1st cycle.

**Figure 3 materials-15-04713-f003:**
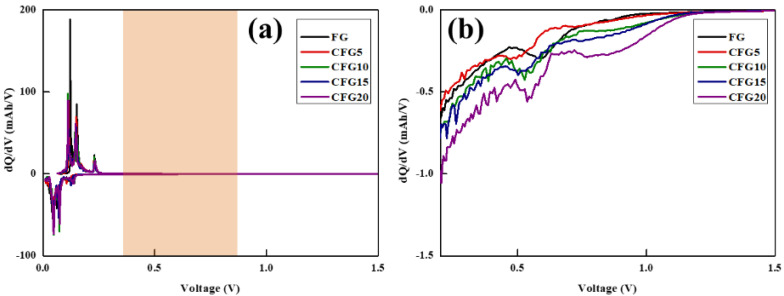
(**a**) Differential capacity (dQ/dV vs. V) curves at 1st cycle and (**b**) more detailed differential capacity of lithiation process.

**Figure 4 materials-15-04713-f004:**
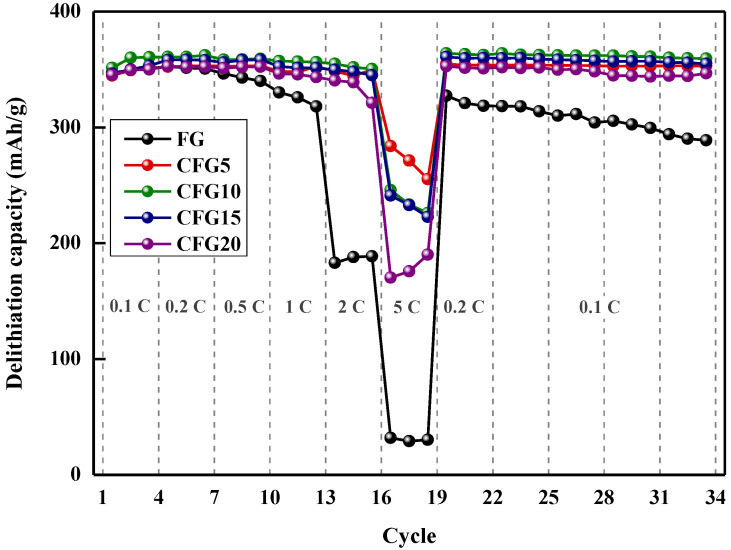
Rate performance at various delithiation rate (1 C rate = 372 mAh/g) and cycling performance at 0.1 C.

**Figure 5 materials-15-04713-f005:**
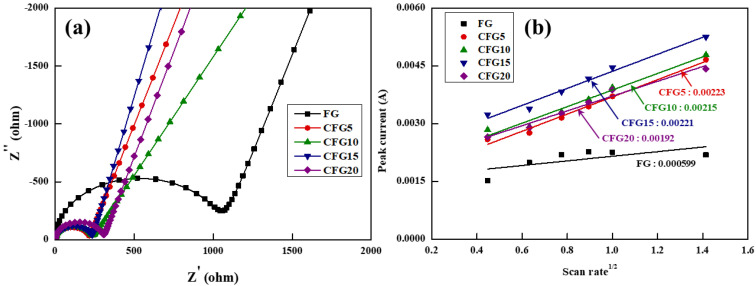
(**a**) Nyquist plot before 1st cycle and (**b**) the relationship between peak current and square root of scan rate based on cyclic voltammetry with various scan rate.

**Figure 6 materials-15-04713-f006:**
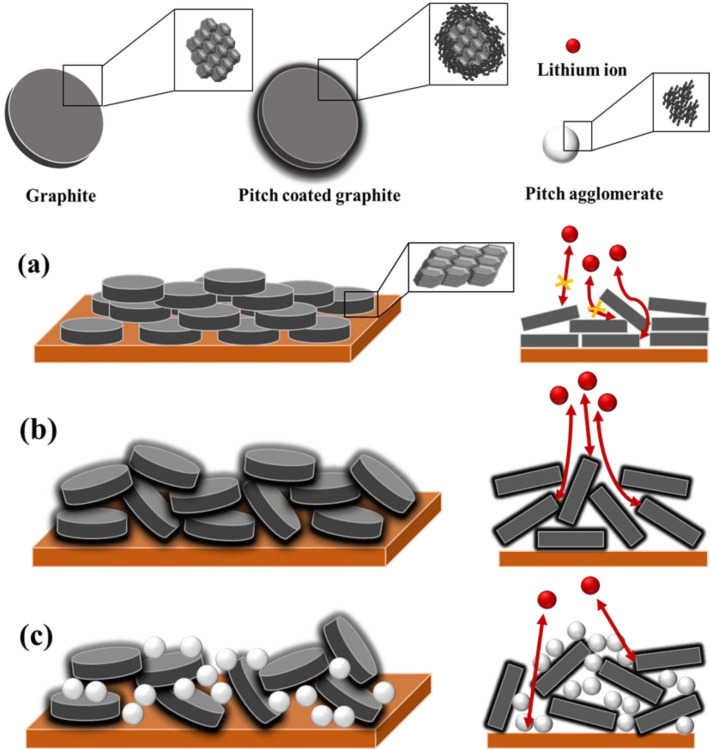
Schematic illustration of electrode structure after pressing process of (**a**) no-coated graphite (FG), (**b**) uniformly pitch-coated graphite (CFG5), and (**c**) excessive pitch-added graphite.

**Table 1 materials-15-04713-t001:** Particle size distribution and surface area to investigate particle aggregation.

	D10	D50	D90	FWHM	BET Surface Area
FG	6.83	14.9	27.43	15.45	32.31
CFG5	8.06	16.65	28.85	15.49	6.36
CFG10	7.97	17.19	29.98	16.54	9.81
CFG15	7.85	18.16	33.16	18.49	5.66
CFG20	5.27	16.76	34.01	21.72	7.06

**Table 2 materials-15-04713-t002:** The A_D_/A_G_ derived from Raman spectroscopy.

	A_D_/A_G_	Average	STDEV
FG	0.585	0.555	0.0452
	0.589		
	0.491		
CFG5	1.837	1.8361	0.0138
	1.852		
	1.819		
CFG10	1.523	1.709	0.1324
	1.821		
	1.783		
CFG15	1.888	1.913	0.1567
	2.117		
	1.736		
CFG20	1.754	1.825	0.2838
	2.203		
	1.519		

**Table 3 materials-15-04713-t003:** Electronic conductivity at a packing density of 1.55.

	FG	CFG5	CFG10	CFG15	CFG20	Pitch (900 °C)
Packing Density (g/cm^3^)	1.55					
Conductivity (S/cm)	188.89	183.16	166.39	161.47	151.73	34.67

## Data Availability

Data available upon request.

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
