# Peer review of "Effect and Mechanism of Pitch Coating on the Rate Performance Improvement of Lithium-Ion Batteries"

_materials, 2022, doi:10.3390/ma15134713_

Round 1

Reviewer 1 Report

The authors have investigated the effect and mechanism of pitch coating on graphite anode in lithium ion batteries. The physical and electrochemical properties of prepared electrodes are also analyzed and compared. However, the following comments are suggested to reply before being published.

1.Will the pyrolyzed pitch coating influence the crystallinity of graphite?

2.Can the author evaluate the content of pyrolyzed pitch coating from CF5 to CF20?

3.CFG5 displays the relative optimal physical and electrochemical performances. The exceeded addition of pitch content will cause the agglomeration of and inferior electrochemical performances of final products. However, the weight ratio of pitch content lower than 5 % is suggested to provide, some relevant literatures (doi:10.1007/s12598-020-01662-4; doi:10.1007/s12598-020-01625-9; Electrochimica, Acta, 2019, 305: 563-570; doi: 10.1016/j.jcis.2021.08.112.) should be referred for the revision.

4.The rate performance of these prepared composites in high current densities with long term cycles are advised to provide to further confirm the rate advantages and cycling performance of CFG5.

5.Can the pitch pyrolyzed amorphous carbon coating influence the density of final composites by controlling the loading of these composites to the same?

Author Response

Thank you for your considerate review, and please see the attachment.

Also, we apologize for the delay in responding to your review report.

We did our best to reply review comments.

Thank you and best regards

Author Response

(The authors gave the same response as above.)

Reviewer 3 Report

High-rate operation of Li-ion cells is hampered by the slow diffusion of lithium ions either through the SEI or the bulk of the (graphite) anode materials. There have been several studies to utilize coke-coated graphite to be able to improve the rate capability and to improve the low temperature performance by being able to use PC-based electrolytes. In this study, the benefits of pitch coating on graphite anode materials and the underlying mechanism are investigated. Evaluation of different proportions of flake pitch mixed with flake graphite reveal that the uniform pitch-coated  graphite, especially with 5% pitch exhibited improved  rate performance owing to the smooth surface with spherical particles. Higher contents of pitch are however not as beneficial because of its aggregation.  This study and the finding are of interest to the battery researchers/technologists and the topic is relevant to the journal. I recommend publishing this manuscript after a revision addressing the following comments:

  1. The discussion is entirely on the slow diffusion of Li-ion through graphite flakes orientated perpendicular to the current flow. In reality, we know that graphite is invariably covered with an SEI and the diffusion of Li-ions through the SEI is more constrained/slower than through bulk of the anode. Any explanation?
  2. Page 2 Line 65: If 5-20% of pitch is mixed with flake graphite, how can it be called a coating (with such large amounts). Also, is there convincing evidence the surface is entirely covered with pitch without any FG on the surface.  How (or why) is pitch so localized on the surface?
  • How is the surface uniformity established, based on the standard deviation in the crystallinity from Raman or is there any other supporting evidence?
  1. What are the respective surface areas of these electrodes with FG and with different blends of pitch? Does the rate performance correlate with the surface area?
  2. It is interesting that with the addition of 5% pitch the surface is populated with spherical particles. But how about layers of FG. How far deep is the spherical particle effect? If the sub-surface layers of FG aren’t spherical, how is the diffusion of Li affected?
  3. Why are the EIS measured at 3V in a fully delithiated (discharge) state, which usually show high impedance values and is not presentative of pattern in the useable voltage range?
  • Wouldn’t it be useful to study the electrode behavior of pitch alone to understand the trends in the rate capability, impedance, and surface spherical particle effect?
  • Page 9: Lines 305-307. Not sure what these couple of sentences mean and why they are here.

Author Response

(The authors gave the same response as above.)

Round 2

Reviewer 2 Report

Having read through the responses to my previously raised review points, I am grateful for the diligent and thorough way that you have worked through and attended to all of the replies. I thank you for the courtesy of taking on board the feedback and hope that it was considered useful by the writing team. but there is still one more minor issue:

1. The BET doesn't decrease with the increased coating content. CFG15 has the lowest BET, but the ICE decrease with the increased coating content. Could the authors please explain why?
